# Proteomic analysis of Masson pine with high resistance to pine wood nematodes

**Jingbin GAO[1,2], Ting PAN[2,3], Xuelian CHEN[2,3], Qiang Wei[2,3], Liuyi Xu[2,3]***

**1** Anhui Vocational & Technical College of Forestry, Hefei, China, **2** State Key Laboratory of the National Forestry and Grassland Administration for Pine Wood Nematode Disease Prevention and Control Technology, Hefei, China, **3** Anhui Academy of Forestry, Hefei, China

\* xly13856910135@163.com

**Data Availability Statement:** The mass spectrometry proteomics data were deposited to the ProteomeXchange (http://proteomecentral. proteomexchange.org/cgi/GetDataset) consortium

## Abstract

Pine wilt disease is a dangerous pine disease globally. We used Masson pine (*Pinus massoniana*) clones, selected through traditional breeding and testing for 20 years, to study the molecular mechanism of their high resistance to pine wood nematodes (PWN, *Bursaphelenchus xylophilus*). Nine strains of seedlings of genetically stable Masson pine screened from different families with high resistance to PWN were used. The same number of sensitive clones were used as susceptible controls. Total proteins were extracted for tandem mass tag (TMT) quantitative proteomic analysis. The key proteins were verified by parallel reaction monitoring (PRM). A threshold of upregulation greater than 1.3-fold or downregulation greater than 0.3-fold was considered significant in highly resistant strains versus sensitive strains. A total of 3491 proteins were identified from the seedling tissues, among which 2783 proteins contained quantitative information. A total of 42 proteins were upregulated and 96 proteins were downregulated in the resistant strains. Functional enrichment analysis found significant differences in the proteins with pectin esterase activity or peroxidase activity. The proteins participating in salicylic acid metabolism, antioxidant stress reaction, polysaccharide degradation, glucose acid ester sheath lipid biosynthesis, and the sugar glycosaminoglycan degradation pathway were also changed significantly. The PRM results showed that pectin acetyl esterase, carbonic anhydrase, peroxidase, and chitinase were significantly downregulated, while aspartic protease was significantly upregulated, which was consistent with the proteomic data. These results suggest that Masson pine can degrade nematode-related proteins by increasing protease to inhibit their infestation, and can enhance the resistance of Masson pine to PWN by downregulating carbon metabolism to limit the carbon available to PWN or for involvement in cell wall components or tissue softening. Most of the downregulated proteins are supposed to act as an alternative mechanism for latter enhancement after pathogen attacks. The highly resistant Masson pine, very likely, harbors multiple pathways, both passive and active, to defend against PWN infestation.

through the PRIDE partner repository with the dataset identifier PXD030664.

**Funding:** Xu L.[T000522] Fifth Batch of Anhui Province Innovation and Entrepreneurship Leading Talent Special Support Program http://www.ahxf. gov.cn/ The funders had no role in study design, data collection and analysis, decision to publish, or preparation of the manuscript. Gao J. [2020dsgzs16] Science Education and Research Project of Education Department of Anhui Province http://jyt.ah.gov.cn/ The funders had no role in study design, data collection and analysis, decision to publish, or preparation of the manuscript.

**Competing interests:** The authors have declared that no competing interests exist.

## Introduction

Pine wilt disease (PWD) is a dangerous pine disease worldwide [1], occurring in many countries. Since it spread to China, it has infested many species of *Pinus*. Initially, it affected black pine (*Pinus thunbergii*), and it now mainly affects Masson pine (*P. massoniana*), which is a pioneer tree species for the greening of barren hills. Masson pine plays an important role in maintaining the safety of the ecological environment and natural landscape, and in soil and water conservation [2].

At present, PWD is mainly prevented by the application of quarantine and chemical pesticides to control pine wood nematodes (PWNs; *Bursaphelenchus xylophilus*) and to kill *Monochamus alternatus* that is an insect vector. Although these methods can slow the rapid spread of PWD, chemical prevention and control measures cause pollution to the environment, making it very difficult to apply pesticides on a large scale, and the cost of prevention is very high. Tree felling and trapping of *M. alternatus* have also worked well. A slow initial response to the discovery of PWN hindered attempts to limit its outward spread; decisive action is needed in the crucial early stages of an outbreak. Recently, fluopyram was found to have significant nematocidal activity against *B. xylophilus* as a new trunk-injection agent [3]. It is very difficult to avoid the occurrence and emergence of new epidemic areas year after year. Because the pathogenesis of PWD and the complexity of its propagation media are not yet fully understood, there are no particularly effective control measures yet. Therefore, the question of how to effectively prevent and control PWD has become urgent in forestry production and management [2]. The sustainable development policies for forests include new regulations for environmentally friendly and cost-effective PWD prevention and control. The selective breeding of PWN-resistant Masson pine is an optimal choice and has attracted more and more attention [4]. Research on the resistance breeding of black pine (*P. thunbergii*) and red pine (*P. densiflora*) in Japan has shown that the breeding of resistant varieties is a very effective measure in the comprehensive prevention and control of PWD [5].

The occurrence and development of PWD are the result of a combination of multiple biological factors such as pine trees, vector insects, PWN, and related microorganisms (https:// www.bspp.org.uk/wp-content/uploads/2021/01/PWD-final-report-1.1EB.pdf). Scientists have tried to find ways to prevent and control PWD based on the physiology and biochemistry of pine trees [6, 7] and on characteristic studies of PWN [8]. The endophytic and rhizospheric microbial community changes are potentially caused by *B. xylophilus* infection in pines [9]. In contrast, the resident bacterial communities, such as *Stenotrophomonas* and *Bacillus* spp., exhibit significant inhibitory activities against PWN during their developmental stages [10], suggesting their potential as biocontrol agents to combat PWD. More precisely, *Bacillus thuringiensis* JCK-1233 has been found to control PWD by elicitation of a moderate hypersensitive reaction before infection to afford pines greater resistance [11].

Research on resistance mechanisms has been carried out from the perspective of resistance genes, with the goal of finding genes related to resistance [12–14]. Recently, maritime pine (*P. pinaster*) and Yunnan pine (*P. yunnanensis*) were compared in terms of resistance to PWN. The results showed that the response was less complex and involved a smaller number of differentially expressed genes in Yunnan pine, which may be associated with its increased degree of resistance to PWN [15]. In addition to the response to PWN infection [16], the species-specific physiology before the infection is very important in a plant's defense strategy.

In this study, we used seedlings of Masson pine with high resistance to PWN and those from sensitive strains as controls. The resistant clones were obtained via classical screening and breeding methods. The proteomic data were collected and analyzed. The expression of many proteins involved in resistance changed and their involvement in resistance is discussed.

Those results suggest that the highly resistant strains possess a network of multiple resistance pathways.

## Materials and methods

### Biological material, pine wood nematode inoculation, and sampling

In the PWD epidemic and nonepidemic areas in Anhui Province, China, in 2000, we collected the seeds of Masson pine trees from a total of 10 counties (Huangshan, Guangde, Jing, Xuanzhou, Xiuning, Nanqiao, He, Qianshan, Taihu, and Quanjiao) with the permission of the local government. In 2001, the seedlings were sown and cultivated in family groups in Hefei, Anhui (11720′E, 31°92′N; altitude, 35 m above sea level). Half of each family's seedlings were transplanted into plastic greenhouses in the spring of 2003 [17]. PWN KS-3B, the most virulent isolate, was selected from 40 infected dead Masson pine in different locations. After 90 days of inoculation with PWN, the first evaluation was performed in July 2003. Seedlings were inoculated with a suspension in 0.5 mL of water containing a mixed population of 5000 nematodes. *P. taeda* and *P. elliottii* were set as the resistant control species. Healthy seedlings from 44,400 seedlings of 384 families were tested. Three to five seedlings were selected from each family as resistant candidates. A total of 1201 Masson pine resistant candidates from 251 families were selected, and these individuals were planted according to a row spacing of 2 × 3 m. In 2006, the shoots from the resistant candidates were grafted and *P. elliottii* was used as the rootstock. The second and third evaluation were performed for two consecutive years in July 2007 and July 2008 [18]. PWN was inoculated with a suspension in 0.5 mL of water containing a mixed population of 10,000 nematodes in the field and virulence was determined if the selected grafted seedlings were resistant each year.

According to the results of the clone resistance evaluation, the highly resistant single trees from different families were selected as the high-resistance strains, and the nonresistant single trees were used as the susceptible group. Nine strains were randomly selected from each group, and three of them were pooled as one biological sample, with a total of 3 biological replicates.

### Tandem mass tag (TMT) quantitative proteome analysis

**Sample preparation.** Tender branches were taken from individual trees. The pine leaves, leaf sheaths, and periderm were removed from the branches, and the phloem was obtained and quickly frozen in liquid nitrogen for 1 min. The samples were kept on dry ice during transportation, and the samples were stored in a –80˚C freezer for later use.

**Protein extraction and trypsin digestion.** The samples were first ground with liquid nitrogen, and then the powder was transferred to a 5 mL centrifuge tube and sonicated three times on ice using a high-intensity ultrasonic processor (Scientz) in lysis buffer (including 1% TritonX-100, 10 mM dithiothreitol, 1% protease inhibitor cocktail, 50 μM of PR-619, 3 μM of TSA, 50 mM NAM, 2 mM EDTA, and 1% phosphatase inhibitor for phosphorylation). An equal volume of Tris-saturated phenol (pH8.0) was added. The mixture was further vortexed for 5 min. After centrifugation (4˚C, 10 min, 5000 *g*), the upper phenol phase was transferred to a new centrifuge tube. The proteins were precipitated by adding at least four volumes of ammonium-sulfate-saturated methanol and incubated at –20˚C for at least 6 h. After centrifugation at 4˚C for 10 min, the supernatant was discarded. The remaining precipitate was washed with ice-cold methanol, followed by washing three times with ice-cold acetone. The protein was redissolved in 8 M urea and the protein concentration was determined using a BCA kit according to the manufacturer's instructions.

The sample was trypsinized according to the instructions from Jingjie PTM BioLab (Hangzhou) Co. Ltd. Finally, the peptides were desalted using a C18 SPE column.

**TMT labeling.** Tryptic peptides were first dissolved in 0.5 M TEAB. Each peptide channel was labeled with the respective TMT reagent (based on manufacturer's protocol, Thermo Scientific) and incubated for 2 h at room temperature. Five microliters of each sample were pooled, desalted, and analyzed by MS to check the labeling efficiency. After the labeling efficiency check, the samples were quenched by adding 5% hydroxylamine. The pooled samples were then desalted with a Strata X C18 SPE column (Phenomenex) and dried by vacuum centrifugation.

**HPLC fractionation.** The sample was fractionated by high-pH reverse-phase HPLC using Agilent 300 Extend C18 column (5 μm particles, 4.6 mm ID, 250 mm length). Briefly, the peptides were separated with a gradient of 8%–32% acetonitrile in 10 mM ammonium bicarbonate (pH 9) over 64 min into 54 fractions. The peptides were then combined into nine fractions and dried by vacuum centrifugation.

**LC–MS/MS analysis.** Afterwards, the peptides were dissolved in the mobile phase A of liquid chromatography and then separated using the EASY-nLC 1200 ultra-high performance liquid system. The whole procedure was guided by Jingjie PTM BioLab (Hangzhou) Co. Ltd. The data acquisition mode uses the data-dependent scanning (DDA) program. In order to improve the effective utilization of the mass spectrometer, the automatic gain control (AGC) was set to 5E4, the signal threshold was set to 2.5 E5 ions/s, the maximum injection time was set to 40 ms, and the dynamic rejection time of the tandem mass spectrometry scanning was set to 30 s to avoid repeated scanning of precursor ions.

**Database search and bioinformatics analysis.** The database used was *P. Massoniana* (14,494 sequences), and a decoy database was added to calculate the false discovery rate (FDR) caused by random matching, while a common contamination database was added to the database to eliminate the influence of contaminated proteins in the identification results.

Protein annotation was carried out using Gene Ontology (GO) analysis, protein structural domain annotation, KEGG pathway annotation, and the subcellular location method, while cluster analysis based on the functional enrichment of proteins and protein interaction network-based analysis was carried out to determine the functional enrichment of proteins using GO enrichment analysis, pathway enrichment analysis, and protein structural domain enrichment analysis.

## Quantitative verification of targeted proteome by mass spectrometry-based parallel reaction monitoring (PRM)

**LC–MS analysis.** After the peptides were separated by the ultra-high-performance liquid system, they were injected into the NSI ion source for ionization and then entered into the Q ExactiveTM Plus mass spectrometer for analysis. The ion source voltage was set to 2.1 kV, and both the peptide precursor ions and their secondary fragments were detected and analyzed using the high-resolution Orbitrap (Jingjie PTM BioLab (Hangzhou) Co. Ltd).

**Proteomic statistical analysis of the data.** First, the average quantitative value was calculated for the three repeats of each group, the resistant or susceptible. The value from the susceptible group was taken as a control. The ratio of the average values between the two groups was calculated and taken as the final differential expression levels. The log2 calculation was performed to gain the relative quantitative values, which conforms to a normal distribution. The two-tailed *t*-test method was used to calculate the $p$-value. A change of differential expression exceeding 1.3 was considered as a significant upregulation when the $p$-value $<0.05$, while a change less than 0.7 was considered as a significant downregulation when the $p$-value $<0.05$.

## Results and discussion

### Evaluation of the resistance of Masson pine clones to PWD

The seeds were collected in 2001 from Masson pine in endemic and nonendemic areas in Anhui provinces, respectively. The seedlings were first inoculated in 2003, and the susceptibility of the seedlings was evaluated 90 days after inoculation (Fig 1A). The second inoculation was carried out for the resistant strains in the same year. The susceptibility of the newly grafted seedlings was evaluated twice after inoculation in 2007 and 2008 (Fig 1B).

### General proteomic analysis

Quantitative proteomics mass spectrometric analysis was carried out for the Masson pine seedlings via TMT labeling, and a total of 342,006 secondary spectrograms were obtained. After the secondary spectrograms of the mass spectrometer were retrieved through the protein theoretical database, the number of available effective spectrograms was 33,220 and the utilization rate of the spectrograms was 9.7%. A total of 18,619 peptides were identified through spectrogram analysis, of which 17,715 were specific peptides. A total of 3491 proteins were identified, of which 2783 could be quantitatively analyzed. The statistical results showed that the biological repeatability between the samples was good enough, particularly for susceptible strains (Fig 2A). An overall distribution diagram of the differential proteins is presented as volcano plots. There were 138 differentially expressed proteins (R/S) obtained in total, of which 42 were upregulated (in red) and 96 were downregulated (in blue in Fig 2B). We then analyzed 17 redundant proteins from each sample for PRM verification and confirmed the data applicability for further analysis (one representative is shown in Fig 2C).

### Functional classification of the differentially expressed proteins

The distribution of the differentially expressed proteins in the secondary annotations of GO is shown in Fig 3 and S1 and S2 Tables. The expression levels of proteins related to cell

A B

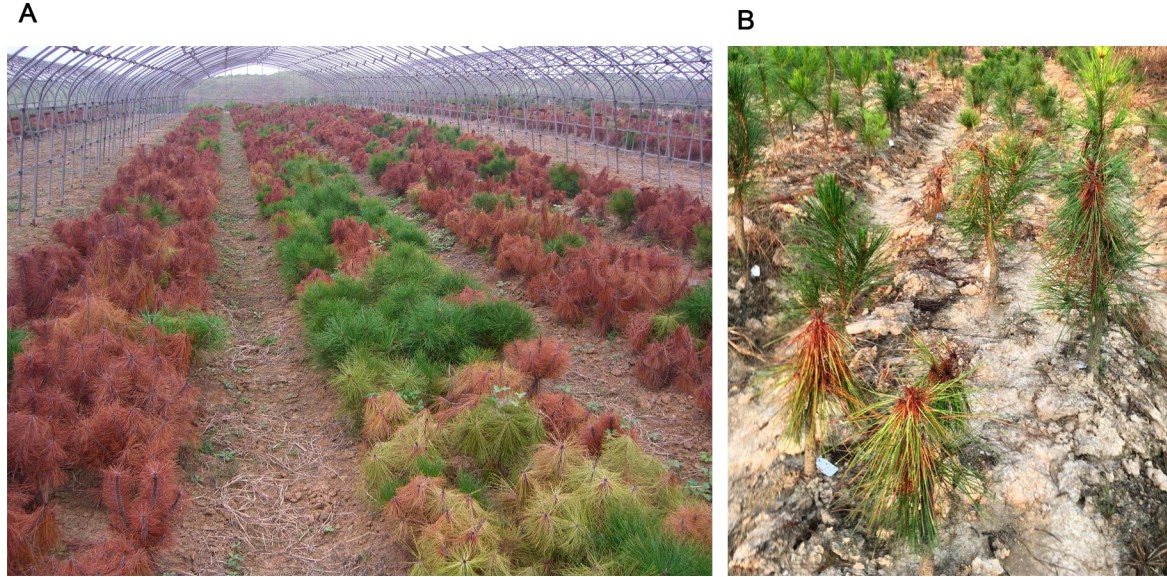

**Fig 1. Resistance screening of Masson pine seedlings 90 days after inoculation with PWN.** (A) The phenotype 90 days post-inoculation of *B. xylophilus* in a plastic greenhouse in 2003. (B) The phenotype 90 days post field inoculation of *B. xylophilus* in 2007. The complete green indicates resistant individuals, while the withered dead plants were considered to be susceptible individuals.

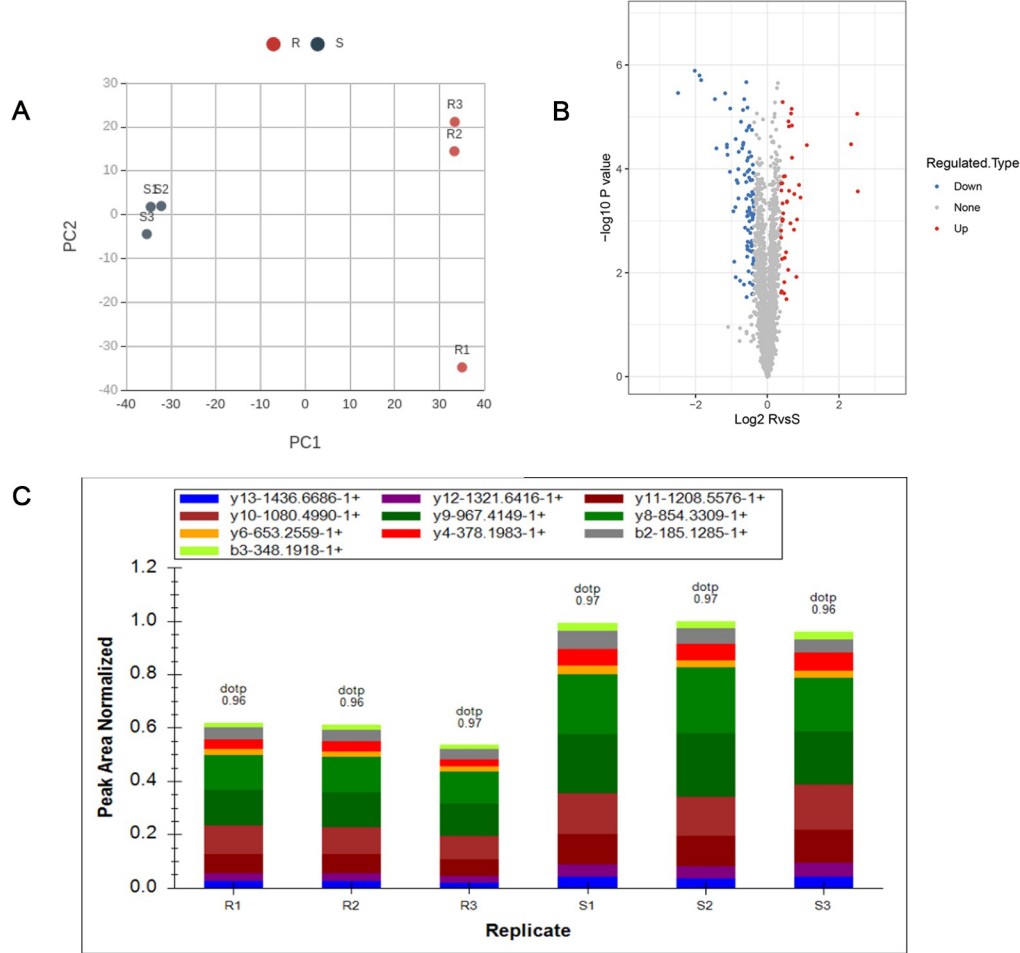

**Fig 2. Repeatability of the samples and overall distribution diagram of the differentially expressed proteins.** (A) Two-dimensional scatter plots for analysis of the protein quantitative principal components among the replicate samples. Three statistical analysis methods, i.e., principal component analysis (PCA), relative standard deviation (RSD), and Pearson's correlation coefficient, were used to evaluate the protein quantitative repeatability. S: susceptible seedlings; R: resistant seedlings. (B) Quantitative volcano plot of the differentially expressed proteins. The horizontal axis is the value of the relative quantitative value of the proteins after log2 logarithmic conversion, and the vertical axis is the value of the significance test p-value after -log10 logarithmic conversion. The red points indicate upregulation, while the blue points indicate downregulation. (C) A representative of parallel reaction monitoring (PRM) verification. A LAYPDIQIISNCDGSSK peak areas graph of the fragment ion was used as an example to confirm the data applicability for further analysis.

metabolism and respondence to external stimuli were very high, and the indexes classified by enzyme activity indicated upregulated and downregulated proteins with catalytic activities (Fig 3A and 3B).

Upon further analysis of the functional enrichment, enzymes such as oxidoreductase, pectinesterase inhibitor, pectinesterase, and carboxylate hydrolase, showed significant differential expression. Moreover, the proteins involved in these biological processes varied substantially not only in expression but also in the number of genes (Fig 4 and S3 Table). The similar results were obtained in agenomic transcriptional study, in which genes encoding pectate lyase were found upregulated in the resistant Masson pine compared with the susceptible clones [13]. It is speculated that changes in these genes might be involved in the resistance to PWD.

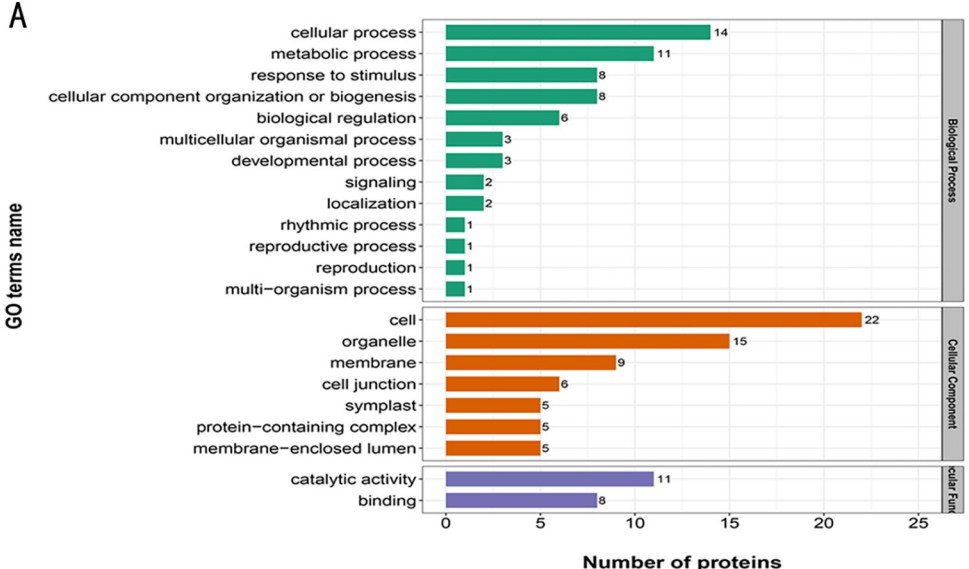

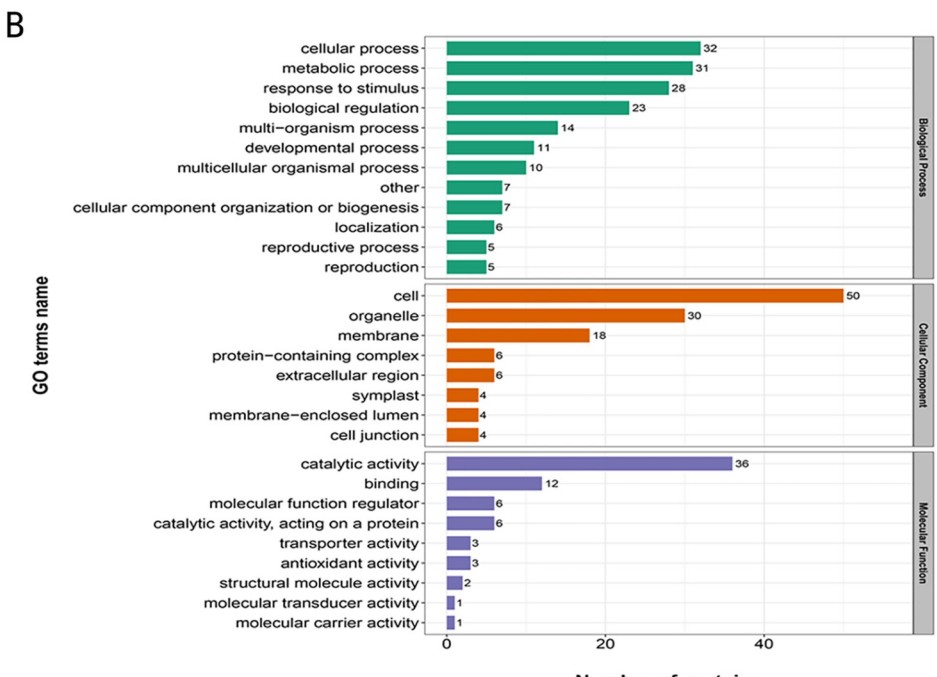

**Fig 3. Distribution of the differentially expressed proteins in the GO secondary annotations and GO-directed acyclic graph in resistant versus susceptible seedlings.** (A) GO functional analysis of proteins with up-regulation in resistant seedlings. (B) GO functional analysis of proteins with down-regulation in resistant seedlings.

## Analysis of the resistance network using the upregulated proteins

Disease resistance in plants depends on genes that are either expressed prior to infection (so-called passive) or induced to express post-infestation by pathogens (so-called active) so that they can defend themselves in a timely manner. Many resistance genes have been characterized

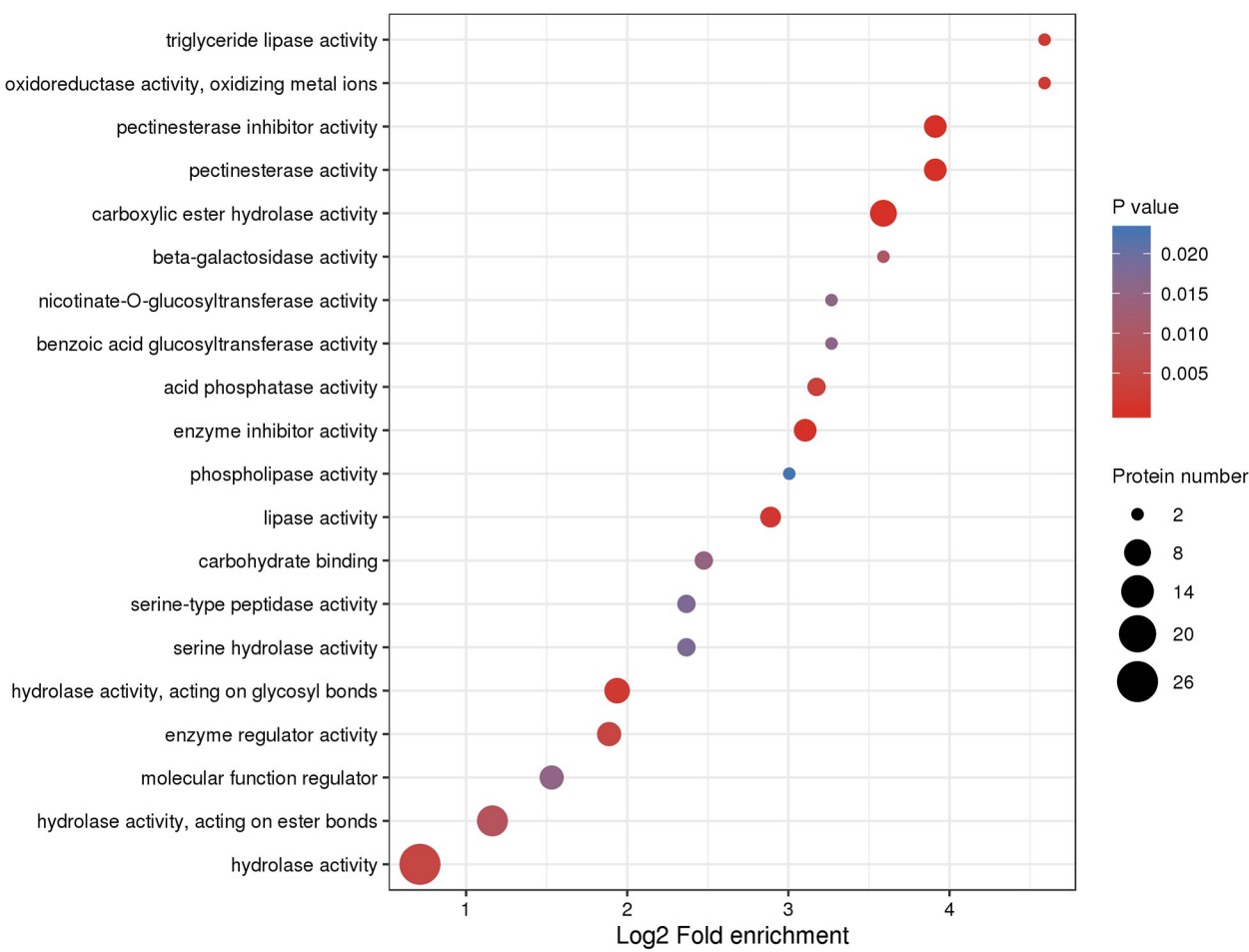

**Fig 4. GO enrichment showing the functions of the differentially expressed proteins.** The size of each circle stands for the number of differentially expressed proteins enriched in corresponding function. The rich factor (log2 fold enrichment) was calculated using the number of enriched genes divided by the number of all background genes in corresponding functions.

in numerous plant species that collectively confer resistance to pathogens, including nematodes and insects [19], as an overall immune system [20]. As shown, we found 42 genes that were upregulated in resistant *P. massoniana* compared to vulnerable clones (S3 Table).

Of these, two aspartic proteinases (At4g16563), a pepsin-like aspartic protease, and aspartic proteinase 4, a saposin-like aspartyl protease family protein, were found to be upregulated by 90% and 39%, respectively, in resistant *P. massoniana*. The enzymes are mostly secreted from cells as inactive proenzymes that activate autocatalytically at an acidic pH. An apoplastic aspartic protease in *Arabidopsis* was identified to contribute to constitutive disease resistance, which is salicylic acid (SA)-dependent [21]. Though SA was not detected in resistant *P. massoniana* strains here, the highly expressed aspartic proteases suggest its big contribution to the defense system against PWN.

Plant nucleotide binding site–Leu-rich repeat (NBS-LRR) proteins are associated with the recognition of and resistance to pathogens, and represent the largest of five known classes of R proteins [22]. In our study, we found two LRR proteins, including plant intracellular Ras-group-related LRR protein 1-like and a Toll/Interleukin-1 receptor-like (TIR)-NBS-LRR, that were upregulated by 34% and 60%, respectively, in resistant *P. massoniana*. PRM verification confirmed the overexpression of LRR proteins, indicating their potential role in resistance to

*B. xylophilus*. In agreement with this view, the downregulation of CC-NBS-LRR in vulnerable strains of *P. massoniana* is involved in the pathogenic pathways of nematode infestation [23]. Serine/arginine-rich (SR) proteins are the key players of alternative splicing, emerging as a critical co-transcriptional regulation for plants in response to environmental stresses [24]. The protein level of this splicing factor RSZ21A increased 44% in our resistant *P. massoniana* strains. This regulation seems to be common in plants and humans in response to pathogen or stress [25, 26].

PLAs (phospholipase A) function to release fatty acids from membrane lipids. In general, phospholipase genes are beneficial for crop yield due to an associated increase in resistance to a variety of pathogen infections and abiotic stresses (reviewed in [27]). Here, a huge increase in PLA1-Iγ3 (>5-fold) was observed in the resistant *P. massoniana*. PLAI in *Arabidopsis* (AtPLAI), an acyl hydrolase that is different from other specific phospholipase A group members, is involved in basal jasmonic acid (JA) production for resistance against the necrotrophic fungus *B. cinerea* [28]. It is likely that JA production in our resistant Masson pine contributes highly to this resistance. Another important, and the most upregulated, protein in the resistant *P. massoniana* is a putative peroxisomal biogenesis factor (peroxin). Peroxisomes are small, membrane-enclosed organelles that contain enzymes involved in a variety of biochemical pathways in different types of cells. In plants, fatty acid oxidation is restricted to peroxisomes, which makes them a major source of metabolic energy. Increased peroxin might be needed to produce more peroxisomes. Thus, enhanced and/or induced peroxisome synthesis may trigger grave consequences for cell fate such as malignant degeneration, but may also rescue cells or tissues from the damaging effects of such radicals [29]. Thus, the enhanced peroxisomes would benefit *P. massonina* to fight *B. xylophilus* infestation and to provide metabolites in a variety of physiological processes.

Terpene synthases were often found to contribute to defense against *B. xylophilus* in *P. massoniana* [4, 6, 13] and another *Pinus Spp.* [15]. In this study, the expression of a few monoterpene synthases was found increased, but not significantly to meet the criteria of the greater-than 30% cutoff, possibly due to no infestation before sampling.

Taken together, these findings suggest that the highly resistant *P. Massoniana* might possess a few mechanisms for resistance to *B. xylophilus* infection and even to other unrelated pathogens. Detailed characterization of the molecular signals that specify the expression of resistance may lead to novel strategies for the protection of *P. massoniana* against PWD.

## Analysis of the resistance network using the downregulated proteins

Interestingly, within the 96 proteins with reduced expression, there were four isoperoxidases (POXs; a decrease of 51%, 33%, 33%, and 23%, respectively) and two laccases (a decrease of 54% and 32%). POXs and laccases are frequently associated with plant defense, which is enhanced after a pathogen attack. Physiologically, POXs also play diverse important roles in the plant life cycle, such as in cell wall metabolism, lignification, suberization, and ROS metabolism. The reason why the seedlings of resistant clones expressed significantly lower levels of peroxidases and laccases than susceptible strains without infestation is elusive. Presumably, these reduced POXs act as an alternative enhanced defense mechanism after pathogen attacks as observed in another study [13] for greater production of phenolic compounds, including synthesis of quinones, tannins, and melanins, and for the polymerization of lignin and suberin. Tannins and melanins are toxic to pathogens, and lignin and suberin are involved in structural defenses. This induction strategy is often used by many species (reviewed in [30]). Another explanation could be the high level of $H_2O_2$ in the resistant *P. massoniana* as a passive defense, as per the previously reported negative correlation between the levels of peroxidases and $H_2O_2$

in the special zone of zucchini (*Cucurbita pepo*) [31]. However, more superoxide radical ($O_2^-$) and $H_2O_2$ accumulated in the susceptible Masson pine after inoculation with PWN and maintained constant under no PWN challenge condition [13]. The definition of the importance of individual POXs could be difficult because of their low substrate specificity *in vitro* and the presence of many isoenzymes. The activity and protein level quantification of each POX after induction of injury by infestation, hurt, or other stresses would provide some hints.

Xyloglucan endotransglucosylase/hydrolase (XTHs; EC 2.4.1.207 and/or EC 3.2.1.151), a xyloglucan-modifying enzyme, has been proposed to have an important role during fruit ripening in the decomposition of polymers of the cell wall [32]. Pectic and hemicellulosic polysaccharides, two of the major cell wall components, would not be solubilized and depolymerized if XTHs expression were reduced. In this study, we found that XTHs expression was lessened by 33% in resistant strains compared to vulnerable strains. And three pectinesterases decreased by 46%, 27%, and 26%. Moreover, one pectin acetylesterase (PAEs) decrease by 63% and one putative pectin methylesterase (fragment) (PME), by 43%. Some galacturonyl residues in pectin are often O-acetylated, which may lift the digestibility of pectin. Therefore, a reduction in PAEs likely protects pine from infestation by *B. xylophilus*. PAEs are widespread in the plant pathogenic oomycetes [33] that consume host pectin as a carbon source. Similarly, demethylesterified pectin may undergo depolymerization by glycosidases. A reduction in PME indicates the protective potential for resistant *P. massoniana* against pathogens. A direct correlation between wall PME activity and strawberry tissue softening has been established [34]. Interestingly, phytopathogenic fungi and bacteria also produce PME for providing virulence to hosts [35].

Additionally, other proteins such as two lipases and one lipid-binding protein, oxidative stress 3 (putative isoform 1), two L-ascorbate oxidases, Class IV chitinase Chia4-Pa1, thaumatin-like protein L2, and a few glycosyltransferases were all expressed significantly less in resistant strains than in susceptible strains. The results are exceptionally unexpected because these gene products are mostly resistance genes in different species [36–40]. As discussed earlier, the activation of genetic and metabolic defenses of plants when facing pathogenic invaders could be gained. These low-expressed proteins would be induced when facing pathogenic attack [38], probably in case of *B. xylophilus* attack on *P. massoniana*. This occurs in Norway spruce (*Picea abies*) after inoculation with *Heterobasidion annosum*, and this infection has been demonstrated to massively and quickly upregulate chitinase IV to reduce or prevent pathogen colonization [41]. A similar result indicates that the *Chia4-Pa* gene is predominantly expressed for resistance in the single-cell zone surrounding the corrosion cavity in Norway spruce seeds [42].

Collectively, the bred highly resistant *P. massoniana* has, very likely, harbored multiple common pathways, both passive and active, to defend against *B. xylophilus* infestation. These findings suggest that Masson pine may be resistant not only to *B. xylophilus*, but also to more unrelated pathogens. Characterization of the molecular signals for intercellular and intracellular barriers as quantitative trait loci (known as QTLs) that specify the expression of resistant genes may lead to more reliable strategies for plant disease control [43].

Though these results are preliminary, they are very intriguing, since the data were collected from highly resistant clones after around 20 years of breeding and field testing. Some more proteins such as low PSII accumulation 1 (chloroplastic isoform X1, encoded by *LPA1*, increase >5-fold), adenosylhomocysteinase (increase by 60%), sucrose synthase (increase by 52%), ferredoxin (reduction to 27.8%), and abscisic acid stress ripening protein homolog (reduction to 46%) are also very interesting (S3 Table). LPA1 appears to be an integral membrane chaperone that is required for efficient PSII assembly, probably through direct interaction with the PSII reaction center protein D1 [44]. *P. massoniana* might own a defense system

against infestation when combined with efficient PSII biogenesis and maintenance, which has enormous economic potential. On the other hand, susceptible clones can be planted in non-infested zones in order to maintain plant community diversity.

## Supporting information

**S1 Table. The protein numbers of GO functional analysis with up-regulation in resistant seedlings.**
(DOCX)

**S2 Table. The protein numbers of GO functional analysis with down-regulation in resistant seedlings.**
(DOCX)

**S3 Table. List of the up-regulated and down-regulated proteins in resistant vs. susceptible strains.**
(DOCX)

## Acknowledgments

The authors greatly thank Drs. Kuanyu Li and Fengmao Cheng for his suggestions during the writing of this manuscript.

## Author Contributions

**Conceptualization:** Jingbin GAO.

**Data curation:** Jingbin GAO, Qiang Wei.

**Formal analysis:** Jingbin GAO.

**Funding acquisition:** Jingbin GAO, Liuyi Xu.

**Investigation:** Jingbin GAO, Xuelian CHEN.

**Methodology:** Jingbin GAO.

**Project administration:** Jingbin GAO.

**Resources:** Xuelian CHEN.

**Software:** Ting PAN.

**Writing – original draft:** Jingbin GAO.

**Writing – review & editing:** Jingbin GAO, Ting PAN.

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
