## [Decision Letter · Decision Letter 0]

29 May 2022

PONE-D-22-06507Proteomic analysis of Masson pine with high resistance to pine wood nematodesPLOS ONE

Dear Dr. Xu,

Thank you for submitting your manuscript to PLOS ONE. After careful consideration, we feel that it has merit but does not fully meet PLOS ONE’s publication criteria as it currently stands. Therefore, we invite you to submit a revised version of the manuscript that addresses the points raised during the review process.

We look forward to receiving your revised manuscript.

Kind regards,

Jon M. Jacobs, Ph.D.

Academic Editor

PLOS ONE

Journal Requirements:

Reviewers' comments:

Reviewer's Responses to Questions

**Comments to the Author**

1. Is the manuscript technically sound, and do the data support the conclusions?

Reviewer #1: Partly

Reviewer #2: Partly

2. Has the statistical analysis been performed appropriately and rigorously? 

Reviewer #1: Yes

Reviewer #2: I Don't Know

3. Have the authors made all data underlying the findings in their manuscript fully available?

Reviewer #1: No

Reviewer #2: No

4. Is the manuscript presented in an intelligible fashion and written in standard English?

Reviewer #1: Yes

Reviewer #2: Yes

5. Review Comments to the Author

Reviewer #1: The authors present a proteomics study in two groups of Masson pine samples with contrasting response to PWN. The used biological material came from individuals selected and tested for 20 years regarding their resistance to PWN. Several putatively important proteins associated to resistance are identified and discussed. This is an important topic given the damaging potential of PWD on pine forests worldwide and the results presented are relevant for PWD research.

However, several major issues should be taken into consideration by the authors to improve the manuscript. These include (1) a more detailed description of the biological materials used and sampling, (2) clarification of the PWD resistance evaluation procedures of the Masson pine clones, and a (3) more focused comparative discussion of the results with previously published work on PWD resistance in pines, as follows:

(1) It is not clear why were 9 lines (what is the definition of a line?) selected, does this mean that each biological sample is a mixture of 3 individual trees? Are these 9 lines from the same or different families? Was the PWN inoculum prepared from several isolates? How were the symptoms evaluated? How long after inoculation was the sampling performed?

(2) The results from the inoculation and the evaluation of resistance are not quantitatively described. If this has not been previously published (in which case a reference should be provided) further detail is needed.

(3) the results from differentially expressed proteins identified should be discussed within the context of previously published results, either using similar comparative approaches, either proteomic or transcriptomic, within the same pine species between resistant and susceptible individuals, e.g. P. pinaster. Such discussion can help to clarify if resistance mechanisms may be similar or specific.

In addition to these major concerns, additional comments are listed below:

Abstract - Please rephrase “Most of the downregulated proteins seemed to take back seats prior to pathogen attacks”

Introduction – the sentence “The endophytic and rhizospheric microbial community

changes that are potentially caused by B. xylophilus infection of pines [9]” seems to be truncated

Results and Discussion – Fig. 1. What is meant by field vaccination?

I could not have access to the data by following the provided link. A supplementary table with detailed info about the proteins and their classification in GO terms (Fig 3) is missing.

The authors claim they found 42 resistant genes that were upregulated in resistant

P. massoniana compared to vulnerable clones. What do the authors mean by resistance genes? Did they use any available database such as Plant Resistance Genes database (Osuna-Cruz et al., 2017)?

It is mentioned that “Though SA was not detected in resistant P. massoniana lines here, the highly expressed aspartic proteases suggest a stronger defense system.” Stronger that what? Please clarify.

Please rephrase the sentence “Thus, tremendously enhanced peroxisomes

would benefit P. massonina as an “evil” to fight B. xylophilus infestation and as a “good” to provide metabolites in a variety of physiological processes.”

Reviewer #2: This manuscript reports on the analysis of proteomic changes in Pinus massoniana resistant and susceptible plants after inoculation with pinewood nematode, the causal agent of pine wilt disease. This manuscript is relevant to understanding better the mechanisms of P. massoniana resistance to this parasite and can be interesting for researchers working on pine wilt disease. However, the manuscript needs improvement before it can be accepted for publishing, especially in the discussion and conclusions.

Throughout the manuscript it is stated that P. massoniana resistant plants “evolved” certain mechanisms of resistance. However, pinewood nematode was introduced recently (40 years ago) in China, giving no time for a long lived tree to evolve resistance to the disease. Instead, resistance is likely to occur due to natural standing variation, and has probably evolved though other selective pressures or genetic drift. Please remove the references to evolution throughout the manuscript.

Introduction:

Line 44: The first time Monochamus alternatus is mentioned there is no explanation why it is important for pine wilt disease, which is confusing. It should be mentioned before that it is the insect vector.

Lines 72-75: Why give details about P. yunnanensis resistance when there is a paper about P. massoniana resistance? Should this not be the focus?

Methods:

In the Methods, more details should be given. Were there any controls inoculated with water? What is/are the sampling timepoint/s? What is the comparison made to obtained differential expression results? Resistant vs control? Resistant vs susceptible? When it says up- or downregulated, what is compared to what? If the comparison is between susceptible and resistant it would be clearer to talk about proteins more expressed in one or the other.

Lines 101-103: This is not very clear. Were the samples pooled in groups of 3 for protein extraction? Needs to be rephrased.

Line 107: Why use only the phloem?

Lines 163-165: This sentence needs more details. “multiple repetitions of each sample” – How many repetitions? And what part of the protocol was repeated?; “average values between the two samples” – What two samples?; “differential expression of the control group” – What control group?

Lines 165-171: What is the program and version used for the statistical analysis?

Results and discussion:

Discussion is confusing in the way it is presented. Why not compare the proteomics results with transcriptomics available for several species after pinewood nematode inoculation, and specifically for P. massoniana resistant varieties (Liu et al. 2017, Scientific Reports)?

Lines 175-179: There should be some results presented here about the inoculation assays, like percentage of plants that are resistant, the differences between families, the disease progression during the time of the experiments, etc. Has this part of the work never been published before? If it was, please provide reference(s).

Fig. 1: “vaccination”?

Lines 215-216: There is no table or image representing expression levels of the proteins. I suggest it is added, even if it is as supplemental material.

Fig. 3: The font is too small, it is difficult to read. (C) What is log2 fold enrichment? Is it a GO enrichment analysis? please make the legend of the image more clear

Lines 238-239: What are resistant genes? Do you mean candidate genes for resistance?

Lines 252-254: This sentence is not clear. Is it a comparison with results obtained elsewhere?

Lines 254-258: I do not see the connection of resistance genes/receptors to alternative splicing? The discussion about the possible role of alternative splicing in resistance is also not clear.

Lines 275-276: It is not clear what you want to say here.

Lines 292-294: This is not supported by what was observed in Liu et al. (2017, Scientific Reports), in which values of H2O2 were similar between resistant and susceptible P. massoniana before pinewood nematode inoculation.

Lines 307-310: Susceptibility to what?

Lines 316-319: This is the opposite of what was found in some transcriptomics papers of pine response to pinewood nematode. This difference should be discussed.

Lines 319-326: I do not understand this part of the discussion and how it relates to your results.

Conclusions:

The text is confusing and it seems like the continuation of the discussion, instead of conclusions.

6. PLOS authors have the option to publish the peer review history of their article (what does this mean?). If published, this will include your full peer review and any attached files.

Reviewer #1: No

Reviewer #2: No

---

## [Author Response · Author response to Decision Letter 0]

5 Jul 2022

Dear editorial staffs,

Thanks to the editor and the two reviewers for intensively reviewing our manuscript. It helps a lot to improve our manuscript. We are happy to be given the opportunity to revise the manuscript. We carefully considered the suggestions or comments and answer the questions. During the revision, we have added more supplemental data to support our claim. The point-to-point responses to the comments were prepared. Please see the below.

Responses to Review #1’s Comments

(1) It is not clear why were 9 lines (what is the definition of a line?) selected, does this mean that each biological sample is a mixture of 3 individual trees? Are these 9 lines from the same or different families? Was the PWN inoculum prepared from several isolates? How were the symptoms evaluated? How long after inoculation was the sampling performed?

Answer: Sorry for not clearing the information in the manuscript. We modified some. The 9 lines are 9 individual seedlings from different families. To make it clearer, we changed the word “line” into “strain”. More detail modification or correction sees the manuscript. We quote some here: “PWN KS-3B, the most virulent isolate, was selected from 40 infected dead Masson pine in different locations. After 90 days of inoculation with PWN, the first evaluation was performed in July 2003. Seedlings were inoculated with a suspension in 0.5 mL of water containing a mixed population of 5000 nematodes. P. taeda and P. elliottii were set as the resistant control species. Healthy seedlings from 44,400 seedlings of 384 families were tested. Three to five seedlings were selected from each family as resistant candidates. A total of 1201 Masson pine resistant candidates from 251 families were selected, and these individuals were planted according to a row spacing of 2 × 3 m. In 2006, the shoots from the resistant candidates were grafted and P. elliottii was used as the rootstock. The second and third evaluation were performed for two consecutive years in July 2007 and July 2008. PWN was inoculated with a suspension in 0.5 mL of water containing a mixed population of 10,000 nematodes in the field and virulence was determined if the selected grafted seedlings were resistant each year. 

According to the results of the clone resistance evaluation, the highly resistant single trees from different families were selected as the high-resistance strains, and the nonresistant single trees were used as the susceptible group. Nine strains were randomly selected from each group, and three of them were pooled as one biological sample, with a total of 3 biological replicates.”

(2) The results from the inoculation and the evaluation of resistance are not quantitatively described. If this has not been previously published (in which case a reference should be provided) further detail is needed.

Answer: Yes, we published one paper concerning the inoculation and the evaluation of resistance. Here is the one “Liu-yi Xu, Jing-bin Gao, Jian Zhang, Xue-lian Chen, Chun-wu Jiang. Evaluation of disease resistance of clones of Pinus massoniana to Bursaphelenchus xylophilus. Journal of Anhui Agricultural University. 2011, 38(6): 848-853. (In Chinese).” We added the reference (#18) into the manuscript.

(3) The results from differentially expressed proteins identified should be discussed within the context of previously published results, either using similar comparative approaches, either proteomic or transcriptomic, within the same pine species between resistant and susceptible individuals, e.g. P. pinaster. Such discussion can help to clarify if resistance mechanisms may be similar or specific.

Answer: We added more comparison with more similar works produced using pine species. Please see the tracked manuscript.

In addition to these major concerns, additional comments are listed below:

Abstract - Please rephrase “Most of the downregulated proteins seemed to take back seats prior to pathogen attacks”

 Answer: We rephrase as “Most of the downregulated proteins are supposed to act as an alternative mechanism for latter enhancement after pathogen attacks”.

Introduction – the sentence “The endophytic and rhizospheric microbial community changes that are potentially caused by B. xylophilus infection of pines [9]” seems to be truncated.

Answer: Sorry, it should be “The endophytic and rhizospheric microbial community changes are potentially caused by B. xylophilus infection in pines”.

Results and Discussion – Fig. 1. What is meant by field vaccination?

Answer: It means “the phenotype 90 days post field inoculation”. To avoid the misunderstanding, we change the wording as “The phenotype 90 days post field inoculation of B. xylophilus”.

I could not have access to the data by following the provided link. A supplementary table with detailed info about the proteins and their classification in GO terms (Fig 3) is missing.

Answer: We supplement Table S1- S3 for Fig. 3 and 4, both of which together are original Fig. 3. Please check it.

The authors claim they found 42 resistant genes that were upregulated in resistant P. massoniana compared to vulnerable clones. What do the authors mean by resistance genes? Did they use any available database such as Plant Resistance Genes database (Osuna-Cruz et al., 2017)?

Answer: Sorry, we checked http://prgdb.org/prgdb4/, any information about pine wilt disease has not been available. We deleted “resistant".

It is mentioned that “Though SA was not detected in resistant P. massoniana strains here, the highly expressed aspartic proteases suggest a stronger defense system.” Stronger that what? Please clarify.

Answer: Sorry for not clarifying it. Now we correct as “…..the highly expressed aspartic proteases suggest its big contribution to the defense system against PWN.”

Please rephrase the sentence “Thus, tremendously enhanced peroxisomeswould benefit P. massonina as an “evil” to fight B. xylophilus infestation and as a “good” to provide metabolites in a variety of physiological processes.”

Answer: Here is the rephrased one: “Thus, the enhanced peroxisomes would benefit P. massonina to fight B. xylophilus infestation and to provide metabolites in a variety of physiological processes”.

Responses to Review #2’s Comments

Throughout the manuscript it is stated that P. massoniana resistant plants “evolved” certain mechanisms of resistance. However, pinewood nematode was introduced recently (40 years ago) in China, giving no time for a long-lived tree to evolve resistance to the disease. Instead, resistance is likely to occur due to natural standing variation, and has probably evolved though other selective pressures or genetic drift. Please remove the references to evolution throughout the manuscript.

Answer: Many thanks to Reviewer 2 for positive acknowledgment. For the “evolve” using, we changed the wording by using “harbor”, “possess”, or “own”. What we expressed is that Masson pine is resistant not only to B. xylophilus, but also to more unrelated pathogens in general. The specificity of any resistance gene needs to be further investigated.

Introduction:

Line 44: The first time Monochamus alternatus is mentioned there is no explanation why it is important for pine wilt disease, which is confusing. It should be mentioned before that it is the insect vector.

Answer: Thanks. We modified as “At present, PWD is mainly prevented by the application of quarantine and chemical pesticides to control pine wood nematodes (PWNs; Bursaphelenchus xylophilus) and to kill Monochamus alternatus that is an insect vector”.

Lines 72-75: Why give details about P. yunnanensis resistance when there is a paper about P. massoniana resistance? Should this not be the focus?

Answer: We referred to it in Introduction to introduce the background in the field.

Methods:

In the Methods, more details should be given. Were there any controls inoculated with water? What is/are the sampling timepoint/s? What is the comparison made to obtained differential expression results? Resistant vs control? Resistant vs susceptible? When it says up- or downregulated, what is compared to what? If the comparison is between susceptible and resistant it would be clearer to talk about proteins more expressed in one or the other.

Answer: Now we gave more detail information as suggested. It seems reviewer #2 misunderstood the materials we described in the manuscript. Sorry for not clarifying well. We did not use water as control, but species P. taeda and P. elliottii were used as positive resistant controls. We screened three times in 2003, 2007, and 2008, respectively. The final resistant materials were used for this study. For proteomic data, it is common to say “upregulation” or “downregulation”. In principle, the expression levels were compared between resistant and susceptible strains.

Lines 101-103: This is not very clear. Were the samples pooled in groups of 3 for protein extraction? Needs to be rephrased.

Answer: We rephrase as “According to the results of the clone resistance evaluation, the highly resistant single trees from different families were selected as the high-resistance strains, and the nonresistant single trees were used as the susceptible group. Nine strains were randomly selected from each group, and three of them were pooled as one biological sample, with a total of 3 biological replicates”.

Line 107: Why use only the phloem?

Answer: We used the phloem because phloem is the first position PWN entering when Monochamus alternatus brings PWN to Masson pine. Definitely, it would be better if we use both the needle and phloem. Sorry, we missed it here. 

Lines 163-165: This sentence needs more details. “multiple repetitions of each sample” – How many repetitions? And what part of the protocol was repeated?; “average values between the two samples” – What two samples?; “differential expression of the control group” – What control group?

Answer: The sampling is actually described in “Biological material, pine wood nematode inoculation, and sampling”. Each group has three biological repeats. Each repeat is a pool of 3 individual seedlings from different families with the similar resistance. We have made it clear in Manuscript. No protocol was repeated. We quote here: “First, the average quantitative value was calculated for the three repeats of each group, the resistant or susceptible. The value from the susceptible group was taken as a control. The ratio of the average values between the two groups was calculated and taken as the final differential expression levels. The log2 calculation was performed to gain the relative quantitative values, which conforms to a normal distribution. The two-tailed t-test method was used to calculate the p-value. A change of differential expression exceeding 1.3 was considered as a significant upregulation when the p-value <0.05, while a change less than 0.7 was considered as a significant downregulation when the p-value <0.05”.

Lines 165-171: What is the program and version used for the statistical analysis?

Answer: Office Excel 2016.

Results and discussion:

Discussion is confusing in the way it is presented. Why not compare the proteomics results with transcriptomics available for several species after pinewood nematode inoculation, and specifically for P. massoniana resistant varieties (Liu et al. 2017, Scientific Reports)?

Answer: Sorry for not comparing more with comparable data in the literature because there are no many data feasible for comparison. Though, we compared more with the paper reviewer#2 mentioned in Results and Discussion (Liu, 2017 ref#13). I quote two here “Similar results were obtained in a genomic transcriptional study, in which genes encoding pectate lyase were found upregulated in the resistant Masson pine compared with the susceptible clones (Liu Q,2017(ref.12).” and “Terpene synthases were often found to contribute to defense against B. xylophilus in P. massoniana (ref. #4,6,13) and another Pinus Spp. (ref#15). In this study, the expression of a few monoterpene synthases was found increased, but not significantly to meet the criteria of the greater-than 30% cutoff, possibly due to no infestation before sampling”.

Lines 175-179: There should be some results presented here about the inoculation assays, like percentage of plants that are resistant, the differences between families, the disease progression during the time of the experiments, etc. Has this part of the work never been published before? If it was, please provide reference(s).

Answer: Yes, a part of the work was published as “Jingbin Gao, Qijun Xi, Zhuyi Sun, TadaoToda. Screening and Breeding of Pinus massoniana Seedlings for Resistant to Pine Wood Nematode. Development of forestry science and technology. 2009, 23(1), 91-95 (ref#17, published in Chinese)”. We referred to it in the manuscript.

Fig. 1: “vaccination”?

Answer: Here is the change in Fig. 1B: “The phenotype 90 days post field inoculation of B. xylophilus” as we answered reviewer #1.

Lines 215-216: There is no table or image representing expression levels of the proteins. I suggest it is added, even if it is as supplemental material.

Answer: All the tables are presented as supplemental data in the revision.

Fig. 3: The font is too small, it is difficult to read. (C) What is log2 fold enrichment? Is it a GO enrichment analysis? please make the legend of the image more clear

Answer: Sorry, the font is really too small. Figure 3 is split into Figure 3 (original Figure 3A and B) and Figure 4 (original Figure 3C) in order to enlarge the font. The current Figure 4 is a GO enrichment analysis. Now we modified the legend and made it clearer.

Lines 238-239: What are resistant genes? Do you mean candidate genes for resistance?

Answer: It was not accurate. Now the “resistant” was deleted in the Revision. 

Lines 252-254: This sentence is not clear. Is it a comparison with results obtained elsewhere?

Answer: Now it is corrected as “In agreement with this view, the downregulation of CC-NBS-LRR in vulnerable strains of P. massoniana is involved in the pathogenic pathways of nematode infestation [ref#23, Xie, 2020]”.

Lines 254-258: I do not see the connection of resistance genes/receptors to alternative splicing? The discussion about the possible role of alternative splicing in resistance is also not clear.

Answer: We did not clearly express our data that the protein levels of serine/arginine-rich splicing factor RSZ21A increased 44% in our resistant Masson pine. Here is the correction: “Serine/arginine-rich (SR) proteins are the key players of alternative splicing, emerging as a critical co-transcriptional regulation for plants in response to environmental stresses [ref#24 Li, 2021]. The protein level of this splicing factor RSZ21A increased 44% in our resistant P. massoniana strains. This regulation seems to be common in plants and humans in response to pathogen or stress [ref#25&26, Wang,2016; Yu,2016]”.

Lines 275-276: It is not clear what you want to say here.

Answer: Here is the correction: “Taken together, these findings suggest that the highly resistant P. massoniana might possess a few mechanisms for resistance to B. xylophilus infection and even to other unrelated pathogens”.

Lines 292-294: This is not supported by what was observed in Liu et al. (2017, Scientific Reports), in which values of H2O2 were similar between resistant and susceptible P. massoniana before pinewood nematode inoculation.

Answer: Yes, we added this information in the Revision. Here is the quotation: “However, superoxide radical (O2−) and H2O2 accumulated more in the susceptible Masson pine after inoculation with PWN, but maintained constant under no PWN-challenge condition (ref#13, Liu, 2017)”. We also cite the paper (ref#13Liu, 2017) concerning the expression of POX as “Presumably, these reduced POXs act as an alternative enhanced defense mechanism after pathogen attacks as observed in another study (ref#13Liu, 2017) for greater production of phenolic compounds…..”.

Lines 307-310: Susceptibility to what?

Answer: Sorry, we deleted. 

Lines 316-319: This is the opposite of what was found in some transcriptomics papers of pine response to pinewood nematode. This difference should be discussed.

Answer: Thanks for reminding. We add more references concerning the opposite results. Here is the quotation: “The results are exceptionally unexpected because these gene products are mostly resistance genes in different species (ref#36-40: PMID: 30709493, 33193547, 34984633, 34681789, 32448419)”.

Lines 319-326: I do not understand this part of the discussion and how it relates to your results.

Answer: The results from original Lines 316-319 are different from other transcriptomic data as reviewer#2 commented. Lines 319-326 are the assumption we proposed and cited some supported evidence.

Conclusions:

The text is confusing and it seems like the continuation of the discussion, instead of conclusions.

Answer: We agree and delete “Conclusion” in Revision.

Thanks to both reviewers and the editor for your comments on our manuscript. We have revised all by point-to-point responses. Hopefully, it fulfills the criteria for publication.

Sincerely,

Jingbin Gao

---

## [Editor Report · Decision Letter 1]

1 Aug 2022

Proteomic analysis of Masson pine with high resistance to pine wood nematodes

PONE-D-22-06507R1

Dear Dr. Xu,

We’re pleased to inform you that your manuscript has been judged scientifically suitable for publication and will be formally accepted for publication once it meets all outstanding technical requirements.

Kind regards,

Jon M. Jacobs, Ph.D.

Academic Editor

PLOS ONE
---

## [Editor Report · Acceptance letter]

4 Aug 2022

PONE-D-22-06507R1 

Proteomic analysis of Masson pine with high resistance to pine wood nematodes 

Dear Dr. Xu:

I'm pleased to inform you that your manuscript has been deemed suitable for publication in PLOS ONE. Congratulations! Your manuscript is now with our production department. 

Kind regards, 

on behalf of

Dr Jon M. Jacobs 

Academic Editor

PLOS ONE